# $\mathcal{ZS}^2$: Zero-Shot Video Sampling from Image

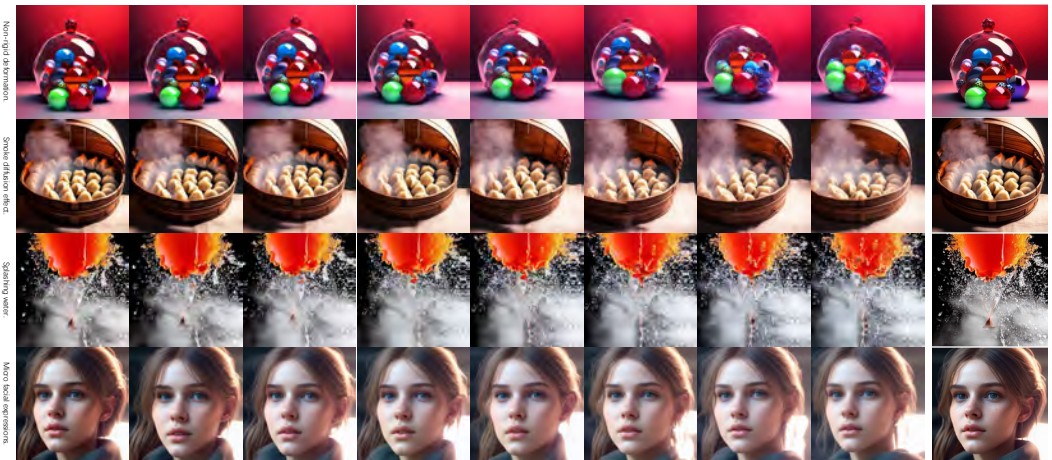

Figure 1: $\mathcal{ZS}^2$ is capable of sampling more detailed and semantically rich motion variations.

## ABSTRACT

Incorporating a temporal dimension into pretrained image diffusion models for video generation is a prevalent approach. However, this method is computationally demanding and necessitates large-scale video datasets. More critically, the heterogeneity between image and video datasets often results in catastrophic forgetting of the image expertise. Recent attempts to directly extract video snippets from image diffusion models have somewhat mitigated these problems. Nevertheless, these methods can only generate brief video clips with simple movements and fail to capture fine-grained motion or non-grid deformation. In this paper, we propose a novel Zero-Shot video Sampling algorithm, denoted as $\mathcal{ZS}^2$, capable of directly sampling high-quality video clips from existing image synthesis methods, such as Stable Diffusion, without any training or optimization. Specifically, $\mathcal{ZS}^2$ utilizes the dependency noise model and temporal momentum attention to ensure content consistency and animation coherence, respectively. This ability enables it to excel in related tasks, such as conditional and context-specialized video generation and instruction-guided video editing. Experimental results demonstrate that $\mathcal{ZS}^2$ achieves state-of-the-art performance in zero-shot video generation, occasionally outperforming recent supervised methods.

## 1 INTRODUCTION

Generative AI has recently attracted substantial attention in the computer vision domain, especially with the advent of diffusion models (Sohl-Dickstein et al., 2015; Ho et al., 2020; Song et al., 2020a;c). These models have demonstrated remarkable efficacy in generating high-quality images from textual prompts, a process referred to as text-to-image synthesis (Ramesh et al., 2022; Rombach et al., 2022; Saharia et al., 2022; Gafni et al., 2022; Xu et al., 2022).

Attempts have been made to extrapolate this success to video generation and editing tasks (Ho et al., 2022b; Singer et al., 2022; Ho et al., 2022a; Wu et al., 2022b; Esser et al., 2023; Molad et al., 2023).

This is achieved by integrating a temporal dimension into the existing image diffusion models. Despite the encouraging outcomes, these methods generally necessitate extensive training with a large corpus of image and/or video data. This requirement can be prohibitively costly and impractical for many users. Moreover, the disparity in training data between image and video datasets often leads to catastrophic forgetting of the image expert (Li & Hoiem, 2018).

To mitigate the cost issue associated with video generation, Tune-A-Video (Wu et al., 2022b) introduced a mechanism that adapts the Stable Diffusion model (Rombach et al., 2022) to the video domain. This strategy significantly curtails the training effort to tuning a single video. However, the generative capabilities of Tune-A-Video are limited to text-guided video editing applications, rendering video synthesis from scratch unachievable.

Recently, Text2Video Zero (Khachatryan et al., 2023) and FateZero (Qi et al., 2023a) have made progress in exploring the novel problem of zero-shot, "training-free" video synthesis. This task involves generating videos from textual prompts without the necessity for any optimization or fine-tuning. By utilizing pre-trained text-to-image models, it capitalizes on their superior image generation quality and extends their applicability to the video domain without additional training. However, these methods primarily generate brief video clips, typically consisting of a few frames, and lack effective control over content, particularly in terms of motion speed.

The fundamental premise of this work is the observation that *continuous video sequences often exhibit substantial correlations within the noise (latent) space* (Ge et al., 2023; Luo et al., 2023). In light of this, we propose an innovative noise initialization model, named the dependency noise model, which supersedes the traditional random noise initialization. To further enhance the continuity of sampled video content over longer segments, we incorporate a temporal momentum mechanism within the self-attention function. The amalgamation of these two techniques gives rise to a new sampling method, denoted as **Zero-Shot video Sampling** or $\mathcal{ZS}^2$ in brief. In comparison to image sampling algorithms such as DDIM (Song et al., 2020a), our proposed $\mathcal{ZS}^2$ video sampling algorithm incurs negligible additional computational overhead. It is straightforward to implement and can be effectively integrated with various sampling algorithms to produce satisfactory video segments. Furthermore, the applicability of our method extends beyond text-to-video synthesis, covering conditional and specialized video generation, as well as instruction-guided video editing.

Our contributions can be encapsulated into the following three aspects:

- We propose a novel zero-shot video sampling algorithm that enables the direct sampling of high-quality video segments from pretrained image diffusion models.

- We present a dependency noise model and temporal momentum attention, which, for the first time, allow us to flexibly control the temporal variations in the generated videos.

- We demonstrate the effectiveness of our method through a broad spectrum of applications, including conditional and specialized video generation, as well as video editing guided by textual instructions.

## 2 BACKGROUND

Present text-to-video synthesis techniques either require costly training on large-scale text-video paired data (Bain et al., 2021), ranging from 1 million to 100 million data points (Wang et al., 2023) or necessitate fine-tuning on a reference video (Wu et al., 2022b). Our objective is to streamline and minimize the cost of video generation by approaching it from a zero-shot video synthesis perspective.

Formally, given a text description $\tau$ and a positive integer $m \in \mathbb{N}$, our aim is to construct a function $\mathcal{F}$ that generates video frames $\mathcal{V} \in \mathbb{R}^{m \times H \times W \times 3}$ (for a predetermined resolution $H \times W$) that exhibit temporal consistency (Khachatryan et al., 2023). Crucially, the function $\mathcal{F}$ should be determined without the necessity for training or fine-tuning on a video dataset. A zero-shot text-to-video method inherently leverages the quality improvements of text-to-image models, thus avoiding the catastrophic forgetting of the image expert.

In this research, we address the zero-shot text-to-video task by utilizing the text-to-image synthesis capability of Stable Diffusion (SD) (Rombach et al., 2021). Given that our objective is to generate

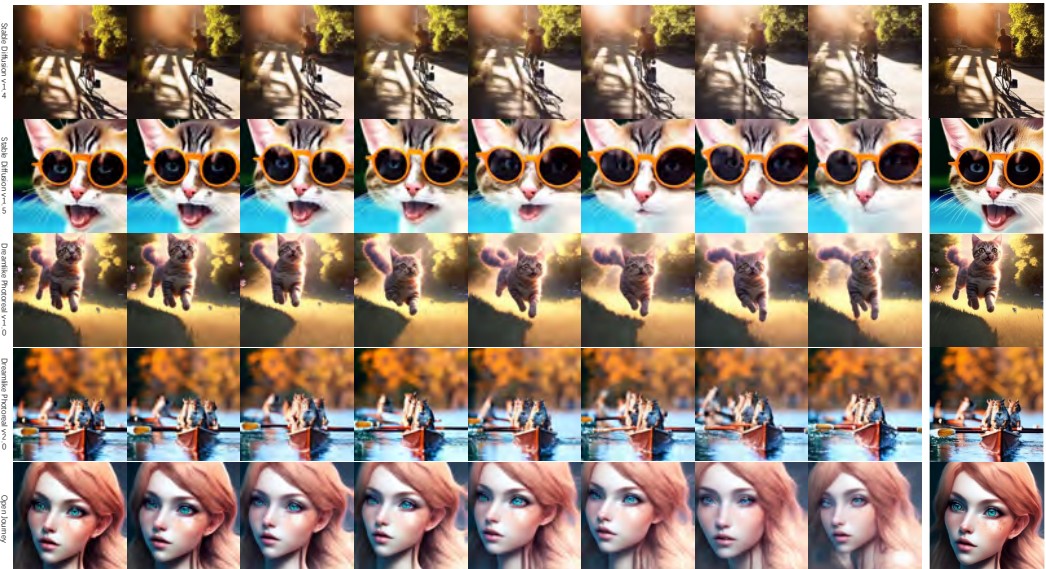

Figure 2: $\mathcal{ZS}^2$ works well across different image diffusion models.

videos rather than images, SD should operate on sequences of latent codes. A direct approach is to independently sample $m$ latent codes from a standard Gaussian distribution, apply DDIM (Song et al., 2020a) sampling to obtain the corresponding tensors $x_0^i$ for $i = 1, \ldots, m$, and then decode to acquire the generated video sequence. However, this results in entirely random image generation that only shares the semantics described by $\tau$, but lacks object appearance and motion coherence.

To overcome these challenges, we propose to (i) incorporate a dependency noise model between adjacent latent codes to ensure consistency in object appearance, and (ii) devise a temporal momentum attention to maintain the motion coherence and identity of the foreground object. Consequently, we construct the **Zero-Shot video Sampling** ($\mathcal{ZS}^2$) algorithm by integrating these two techniques into the DDIM sampling methods, enabling the direct sampling of high-quality videos from SD and other diffusion models. Each component of our proposed method is discussed in detail in the following sections. Preliminary knowledge about the image diffusion model and attention function can be found in Appendix B.

## 3 DEPENDENCY NOISE MODEL

The image diffusion model is trained to eliminate independent noise from a perturbed image. The noise vector $\epsilon$ in the denoising objective is sampled from an i.i.d. Gaussian distribution $\epsilon \sim \mathcal{N}(\mathbf{0}, \mathbf{I})$. However, after training the image diffusion model and applying it to reverse real frames from a video into the noise space on a per-frame basis, the noise maps corresponding to different frames exhibit high correlation (Ge et al., 2023; Luo et al., 2023).

In this study, our goal is to explore the design space of noise priors and propose a model that is optimally suited for our video sampling task, which results in significant performance improvements. We represent the noise corresponding to individual video frames as $\epsilon^1, \epsilon^2, \ldots \epsilon^m$, where $\epsilon^i$ corresponds to the $i^{th}$ element of the noise tensor $\epsilon$. PYoCo (Ge et al., 2023) has developed two intuitive noise models, namely, the mixed and progressive noise model, to introduce correlations among $\epsilon^{1:m}$.

The *Mixed noise model*, also known as the residual noise model or individual noise model, has been utilized in Luo et al. (2023) to expedite the convergence of the video diffusion model. In the mixed noise model, we generate two noise vectors: $\epsilon_{\text{shared}}, \epsilon_{\text{ind}} \sim \mathcal{N}(\mathbf{0}, \mathbf{I})$. $\epsilon_{\text{shared}}$ is a universal noise vector shared across all video frames, while $\epsilon_{\text{ind}}$ is the individual noise per frame. The final noise is a linear combination of these two vectors: $\epsilon^i = \sqrt{\alpha}\epsilon_{\text{shared}} + \sqrt{1-\alpha}\epsilon_{\text{ind}}^i$.

The *Progressive noise model*, also known as the linear noise model, generates noise for each frame in an autoregressive manner, where $\epsilon^i$ is produced by perturbing $\epsilon^{i-1}$. Let $\epsilon^0, \epsilon_{\text{ind}}^i \sim \mathcal{N}(\mathbf{0}, \mathbf{I})$ denote

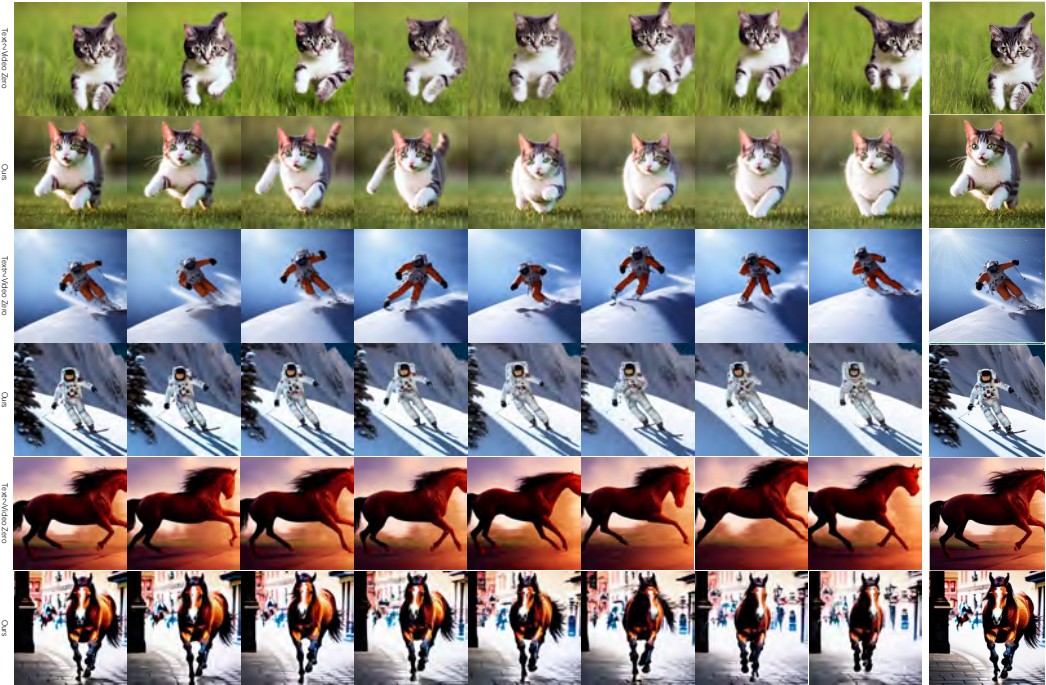

Figure 3: Comparison with baseline: Text2Video-Zero (Khachatryan et al., 2023).(Both sampled from *Dreamlike Photoreal v2.0* (dreamlike art, 2022b))

the independent noise generated for the first frame and $i$th frame. Then, progressive noising can be formulated as: $\epsilon^i = \sqrt{\alpha}\epsilon^{i-1} + \sqrt{1-\alpha}\epsilon^i_{\text{ind}}$.

In both models, the parameter $\alpha$, ranging from 0 to 1, governs the extent of noise shared across different video frames. A larger $\alpha$ signifies a stronger correlation among the noise maps corresponding to various frames. As $\alpha$ approaches 1, all frames are imbued with identical noise, resulting in the creation of a static video. On the contrary, $\alpha = 0$ is indicative of independent and identically distributed (i.i.d.) noise.

The employment of mixed and progressive noise models for the training of a video diffusion model has demonstrated effectiveness, as evidenced in Ge et al. (2023). This approach enables the efficient learning of animation transitions between frames during the training process. However, as illustrated in Figure 14 and 15, despite the strong correlations induced among $\epsilon^{1:m}$ by these two sampling techniques, their direct implementation in zero-shot video sampling is not feasible.

## 3.1 DEPENDENCY NOISE MODEL

To generate more structured noise sequences, $\epsilon^{1:m}$, that encapsulate animation more effectively, we propose a novel dependency noise model. This model employs KL divergence as a regulatory mechanism for the correlation between two successive frames.

Specifically, the model stipulates that for all $\epsilon^i \sim \mathcal{N}(\mathbf{0}, \mathbf{I})$, the KL divergence between $\epsilon^i$ and $\epsilon^{i-1}$ should approximate $\lambda_i$. This requirement necessitates the minimization of the following objective function: $\mathcal{L}(\epsilon^i, \epsilon^{i-1}, \lambda_i) = \| KL(\epsilon^i, \epsilon^{i-1}) - \lambda_i \|_2^2$:

$$\arg\min_{\epsilon^{1:m}} \mathcal{L}(\epsilon^i, \epsilon^{i-1}, \lambda_i), \text{s.t.}, \epsilon^i \sim \mathcal{N}(\mathbf{0}, \mathbf{I}) \tag{1}$$

for $i \in \{2, \ldots, m\}$. Here, $\lambda_i$ serves as a control parameter for the KL divergence between two consecutive frames. By adjusting $\lambda_i$, we can more effectively regulate the rate of content changes between frames. When $\lambda_i \to 0$, all frames incorporate identical noise, resulting in a static video, a situation analogous to that of $\alpha \to 1$. Conversely, $\lambda_i \to \infty$ corresponds to i.i.d. noise.

Revisiting Eq. 1, given $\epsilon^{i-1}$, $\epsilon^i$ can be computed via $\epsilon^{i-1} - \lambda_i / \exp\epsilon^{i-1}$, a derivation that originates from the definition of KL divergence. However, this analytical solution $\epsilon^i$ does not necessarily

---

**Algorithm 1** $\mathcal{ZS}^2$: **Z**ero-**S**hot video **S**ampling Algorithm

---

**Require:** $\text{DM}_{\text{TMA}}$: Diffusion Model with Temporal Momentum Attention. $\mu, \lambda$: hyper-parameters to regulate the dependency noise model and temporal momentum attention, respectively. $m$: length of the video sequence.

1: $\epsilon_T^1 \sim \mathcal{N}(\mathbf{0}, \mathbf{I})$          ▷ Randomly sample the initial latent code.
2: **for** $i = 2$ to $m$ **do**
3:      $\tilde{\epsilon}^i \sim \mathcal{N}(\mathbf{0}, \mathbf{I})$          ▷ Initialize $\tilde{\epsilon}^i$ with random noise.
4:      **repeat**          ▷ Random search phase.
5:          $\epsilon \sim \mathcal{N}(\mathbf{0}, \mathbf{I})$          ▷ Randomly sample a dependent noise.
6:          **if** $\mathcal{L}(\epsilon, \epsilon_T^{i-1}, \lambda_i) \leq \mathcal{L}(\tilde{\epsilon}^i, \epsilon_T^{i-1}, \lambda_i)$ **then**
7:              $\tilde{\epsilon}^i \leftarrow \epsilon$          ▷ Substitute $\tilde{\epsilon}^i$ with $\epsilon$.
8:          **end if**
9:      **until** Limit of iterations is reached.
10:      $\alpha = 0, \delta = 0.1$          ▷ Initialize coefficient $\alpha$ and step size $\delta$.
11:      **repeat**          ▷ Linear search phase.
12:          **if** $\mathcal{L}(\sqrt{\alpha + \delta}\epsilon_T^{i-1} + \sqrt{1 - \alpha - \delta}\tilde{\epsilon}^i, \epsilon_T^{i-1}, \lambda_i)$ decreases **then**
13:              $\alpha \leftarrow \alpha + \delta$          ▷ Increment $\alpha$ by $\delta$.
14:          **else**
15:              $\delta \leftarrow \frac{\delta}{2}$          ▷ Otherwise, try with a smaller step size by halving $\delta$.
16:          **end if**
17:      **until** Convergence is achieved.
18:      $\epsilon_T^i = \sqrt{\alpha}\epsilon_T^{i-1} + \sqrt{1 - \alpha}\tilde{\epsilon}^i$          ▷ Generate $\epsilon_T^i$ using linear interpolation.
19: **end for**
20: **return** $\text{DDIM}(\text{DM}_{\text{TMA}}, \epsilon_T^{1:m}, \mu)$          ▷ Execute the standard DDIM sampling algorithm.

---

adhere consistently to the constraint, i.e., $\epsilon^i \sim \mathcal{N}(\mathbf{0}, \mathbf{I})$. In fact, as the video sequence extends, this analytical solution tends to deviate significantly from the normal distribution, which results in the sampled noise being unable to generate valid content via diffusion models.

As illustrated in Algorithm 1, we propose a two-stage noise search algorithm, which is a departure from the conventional analytical solution.

In the first stage, referred to as the random search stage, we generate a set of independent noises by sampling from the normal distribution $\mathcal{N}(\mathbf{0}, \mathbf{I})$. The noise with the KL divergence closest to $\lambda_i$ when juxtaposed with $\epsilon^{i-1}$ is selected as the initial value of $\epsilon^i$, represented as $\tilde{\epsilon}^i$.

In the subsequent stage, we aim to find a coefficient $\alpha \in [0, 1]$ that results in $\epsilon^i = \sqrt{\alpha}\epsilon^{i-1} + \sqrt{1 - \alpha}\tilde{\epsilon}^i$, thereby minimizing Eq. 1.

As demonstrated in Figure 17, our proposed two-stage algorithm effectively identifies the necessary noise sequence $\epsilon^{1:m}$ within a limited number of iterations. Concurrently, Figure 16 provides evidence that the dependency noise model, to a certain degree, exhibits superior regularity in comparison to the other two noise models.

## 4 TEMPORAL MOMENTUM ATTENTION

To leverage the potential of cross-frame attention and employ a pretrained image diffusion model without necessitating retraining, FateZero (Qi et al., 2023b) replaces each self-attention layer with cross-frame attention. In this setup, the attention for each frame is primarily directed towards the initial frame. A similar structure is also adopted in Khachatryan et al. (2023).

In more detail, within the original UNet architecture $\epsilon_\theta^t(x_t, \tau)$, each self-attention layer receives a feature map $x \in \mathbb{R}^{h \times w \times c}$, which is then linearly projected into query, key, and value features $Q, K, V \in \mathbb{R}^{h \times w \times c}$. The output of the layer is computed using the following formula (for simplicity, this is described for only one attention head) (Vaswani et al., 2017): $\text{SA}(Q, K, V) = \text{Softmax}\left(\frac{QK^T}{\sqrt{c}}\right)V$.

In the context of video sampling, each attention layer is provided with $m$ inputs: $x^{1:m} = [x^1, \ldots, x^m] \in \mathbb{R}^{m \times h \times w \times c}$. As a result, the linear projection layers produce $m$ queries, keys, and values $Q^{1:m}, K^{1:m}, V^{1:m}$, respectively. Therefore, we replace each self-attention layer with cross-frame attention, where each frame's attention is focused on the initial frame, as follows: $\text{CFA}(Q^i, K^{1:m}, V^{1:m}) = \text{Softmax}\left(\frac{Q^i(K^1)^T}{\sqrt{c}}\right) V^1$ for $i = 1, \ldots, m$.

The application of cross-frame attention aids in the transfer of appearance, structure, and identities of objects and backgrounds from the first frame to the subsequent frames. However, this method lacks the connection between adjacent frames, which could lead to significant variations in the generated video sequence, as depicted in Figure 3.

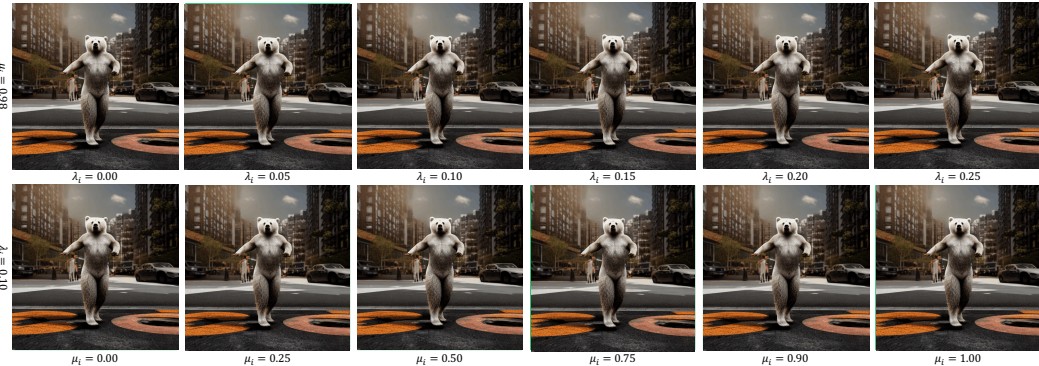

Figure 4: The motion is regulated by $\lambda_i$ and $\mu_i$. We present several video samples from the pose guidance task. From the first and second rows, it is evident that different values of $\lambda_i$ and $\mu_i$ can effectively control the variations in video content. (Best viewed with *Adobe Acrobat*.)

### 4.1 TEMPORAL MOMENTUM ATTENTION

Our observations indicate that self-attention, due to its lack of inter-frame context, results in a more diverse set of sampled features. On the other hand, cross-frame attention relies solely on information from the initial frame. While this ensures the consistency of the sampled results, it also leads to a reduction in diversity.

To strike a balance between the distinct effects of self-attention and cross-frame attention, we introduce Temporal Momentum Attention (TMA). The mathematical representation of TMA is as follows:

$$\text{TMA}(Q^i, K^{1:i}, V^{1:i}) = \text{Softmax}\left(\frac{Q^i(\ddot{K}^{1:i})^T}{\sqrt{c}}\right) \ddot{V}^{1:i} \tag{2}$$

This applies for $i = 1, \ldots, m$, where $\ddot{K}^{1:i} = \mu_i \ddot{K}^{1:i-1} + (1 - \mu_i)K^i$ and $\ddot{K}^{1:1} = K^1$. The same definition applies to $\ddot{V}^{1:i}$.

It's clear that when all $\mu_i$ values are set to 1, TMA becomes equivalent to cross-frame attention. Conversely, when all $\mu_i$ values are set to 0, TMA becomes equivalent to self-attention. As illustrated in Figure 4, by suitably controlling the value of $\mu$, we can generate more optimal video sequences.

**Efficient Computation of $\ddot{K}^{1:i}$.** A straightforward approach to calculate the values of $\ddot{K}^{1:i}$ would involve computing them individually using a for loop. However, to fully leverage the computational capabilities of the GPU, we propose the use of matrix operations to concurrently compute all $\ddot{K}^{1:i}$ values. This method specifically requires the construction of an upper triangular coefficient matrix $U \in \mathbf{R}^{m \times m}$. The vector of $\ddot{K}^{1:i}$ is then obtained through a matrix multiplication operation as follows:

$$\left[\ddot{K}^{1:1}, \ddot{K}^{1:2}, \cdots, \ddot{K}^{1:m}\right] = \left[K^1, (1-\mu)K^2, \cdots, (1-\mu)K^m\right] \underbrace{\begin{bmatrix} \mu^0 & \mu^1 & \cdots & \mu^{m-1} \\ 0 & \mu^0 & \cdots & \mu^{m-2} \\ \vdots & \vdots & \cdots & \vdots \\ 0 & 0 & \cdots & \mu^0 \end{bmatrix}}_{:=U}$$

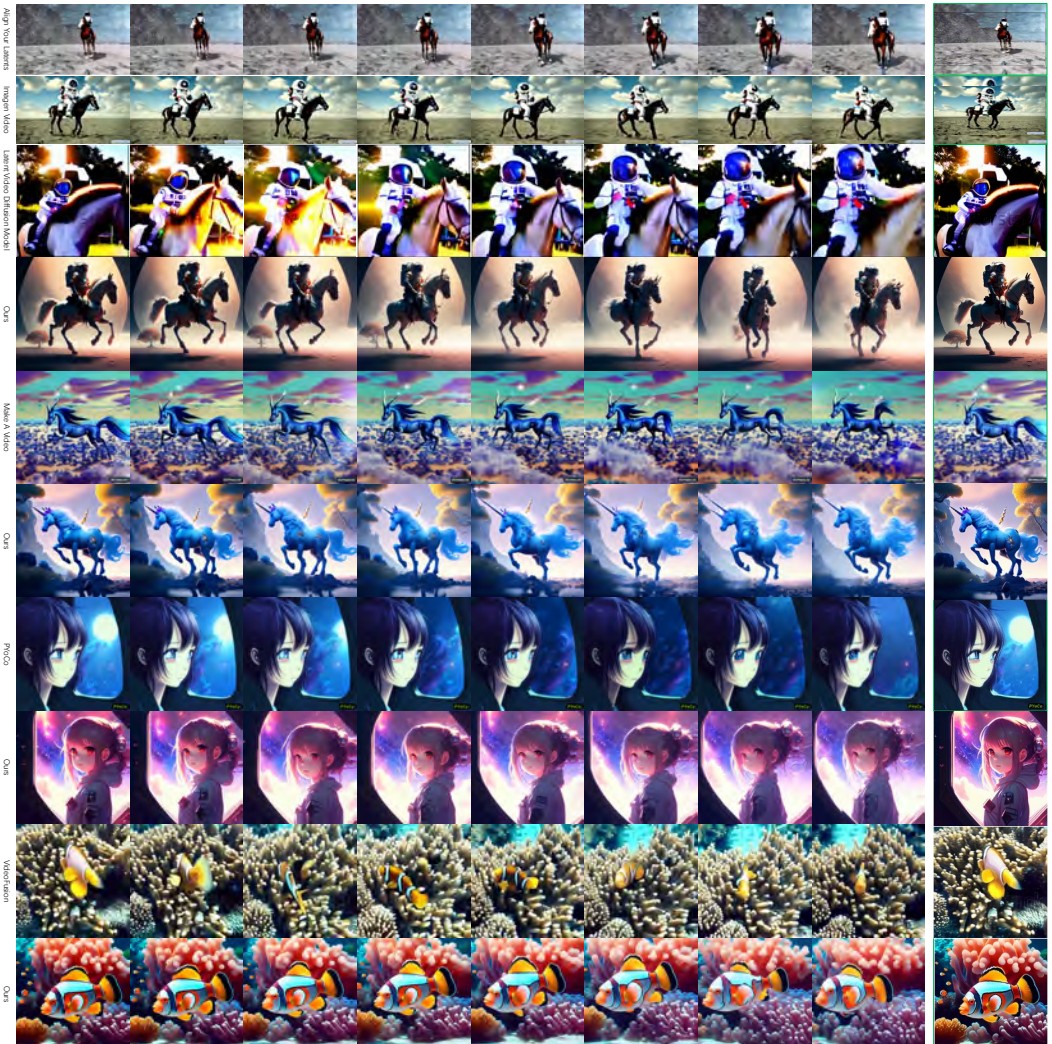

Figure 5: Comparison with supervised video diffusion models. From top to bottom: Align Your Latents (Blattmann et al., 2023), Imagen Video (Ho et al., 2022a), Latent Video Diffusion Model (He et al., 2022), Make A Video (Singer et al., 2022), PYoCo (Ge et al., 2023), VideoFusion (Luo et al., 2023).

In general, when the exponent $i$ of $\mu^i$ is relatively large, $\mu^i$ approaches 0. These elements can be ignored to further reduce computational overhead.

## 5 ZERO-SHOT VIDEO SAMPLING ALGORITHM

By incorporating the dependency noise model and temporal momentum attention, we have successfully sampled high-quality videos from the image diffusion model, leveraging the existing DDIM algorithm. This process is outlined in Algorithm 1. The implementation details of the dependency noise model and temporal momentum attention are provided in Appendix D.

Interestingly, when the video sampling a single image, i.e., $m = 1$, the dependency noise model simplifies to a random noise model, and the temporal momentum attention simplifies to self-attention. This suggests that irrespective of the values assigned to $\lambda_i$ and $\mu_i$, the $\mathcal{ZS}^2$ sampling algorithm will consistently produce results identical to those of the original DDIM algorithm. This feature ensures the high compatibility of the $\mathcal{ZS}^2$ algorithm with various sampling algorithms and coding frameworks, eliminating the need for additional project maintenance.

|  | SDv1.4 (2022a) | SDv1.5 (2022b) | DPv1.0 (2022a) | DPv2.0 (2022b) | OJ (2022) |
|---|---|---|---|---|---|
| DDIM (2020b) | 27.0 | 27.2 | 27.5 | 27.3 | 26.4 |
| Text2Video-Zero(2023) | 26.1 | 26.8 | 27.3 | 27.1 | 25.9 |
| $\mathcal{ZS}^2$(Ours) | 26.9 | 27.0 | 27.7 | 27.2 | 26.4 |

Table 1: Clip score of different sampling methods with different diffusion models. It's important to note that the DDIM only samples one image at a time, while other methods sample $m$ frames each time.

**Comparison with related works.** Text2Video-Zero (Khachatryan et al., 2023) and $\mathcal{ZS}^2$ are contemporaneous works, both aiming to develop innovative sampling methods for zero-shot video generation. However, Text2Video-Zero, to achieve satisfactory sampling results, incorporates motion dynamics in latent codes, necessitating additional DDIM backward and DDPM forward computations. To further ensure the continuity of the video background, it also employs a saliency detection method for background smoothing. This not only escalates the computational overhead but also complicates the algorithm implementation, thereby limiting its flexibility and applicability. In contrast, $\mathcal{ZS}^2$ offers a significant advantage in these aspects. Moreover, our experimental results demonstrate that the video clips sampled by $\mathcal{ZS}^2$ are noticeably superior to those generated by Text2Video-Zero.

# 6 EXPERIMENTS

## 6.1 IMPLEMENTATION DETAILS

Unless otherwise stated, our primary diffusion model is Dreamlike Photoreal v1.0. In this model, all $\lambda_i$ values are set to $0.01$, and all $\mu_i$ values are configured to $0.98$. Algorithm 1 includes a random search phase and a linear search phase, with the iteration count set to $10$ and $15$, respectively. In our experiments, we generate $m = 8$ frames, each with a resolution of $512 \times 512$, for every video clip. However, our framework is inherently flexible and can generate an arbitrary number of frames. This can be achieved either by increasing $m$ or by using our method in an auto-regressive manner, where the last generated frame $m$ is used as the initial frame for computing the subsequent $m$ frames. All prompts used for generating video clips in each figure are provided in Appendix C. We also include an mp4 format video clip in the last column of most figures, which can be viewed using *Adobe Acrobat* [1].

## 6.2 COMPREHENSIVE COMPARISON IN TEXT TO VIDEO TASK

In this study, we provide an extensive comparison between our method and Text2Video-Zero, another zero-shot video synthesis method, from both quantitative and qualitative aspects. You can refer to Appendix E for more sampled video clips with different diffusion models.

From a quantitative standpoint, we utilize the CLIP score (Hessel et al., 2021), a metric for video-text alignment, for evaluation. We randomly select 100 videos generated by both DDIM and Text2Video-Zero using five distinct diffusion models, resulting in a total of 500 videos. We then synthesize corresponding videos using the same prompts as per our method, where DDIM samples $m$ independent images. The CLIP scores are presented in Table 1. Both methods alter the inference and sampling process of the diffusion models, which might introduce unseen noise distributions during training, potentially affecting the sampling quality. However, our method, as indicated by the CLIP scores, yields results more closely aligned with DDIM, thereby showcasing the superiority and generalizability of our approach. Interestingly, we even surpass DDIM in terms of CLIP scores for some diffusion models. We attribute this to $\mathcal{ZS}^2$'s effective utilization of temporal information during the sampling process, which enhances the quality of individual frame sampling.

From a qualitative perspective, we provide visualizations of some generated video clips in Figure 3. Our method's sampled video segments clearly exhibit superior continuity, significantly reducing abrupt video frames. Contrasting with the simple upward and downward object motions in the motion field in Khachatryan et al. (2023), the noise sampled by our dependency noise model can diffuse more specific, complex motions and generalize well across different diffusion models, as depicted in Figure 2. Coupled with temporal momentum attention, our method can generate more intricate mo-

---

[1]https://acrobat.adobe.com/us/en/

tions for more challenging objects, such as fluid's non-rigid deformation, complex smoke diffusion effects, and even subtle facial micro-expressions, as shown in Figure 1.

**Qualitative comparison with supervised video diffusion models** In Figure 5, we present a comparison between short videos generated by $\mathcal{ZS}^2$ and various supervised video diffusion models. It's evident that the video frames sampled by our method generally display superior image quality, while those sampled by the video diffusion model appear noticeably blurred. This discrepancy primarily stems from the lack of video clips (in the order of millions) for training, compared to the image dataset (in the tens of billions) during the training process of the video diffusion models. This inherent data deficiency results in the suboptimal quality of the video diffusion model output. Consequently, a combined training approach of video and image, or training based on a pre-trained image diffusion model is often adopted. However, this method fails to fully exploit the prior knowledge of the image, leading to significant catastrophic forgetting of the image expert as training progresses.

By incorporating a spatio-temporal super-resolution model for post-processing, we can convert the video segments sampled by $\mathcal{ZS}^2$ into high-resolution and more fluid video segments, as illustrated in Figure 6. Our approach of initially sampling video clips via zero-shot, followed by the application of a spatiotemporal super-resolution model for post-processing, effectively bypasses the catastrophic forgetting of the image expert and provides a novel solution for video generation.

## 6.3 EXTENTIONS

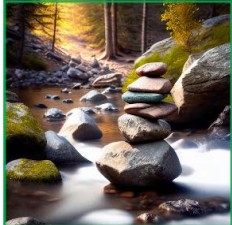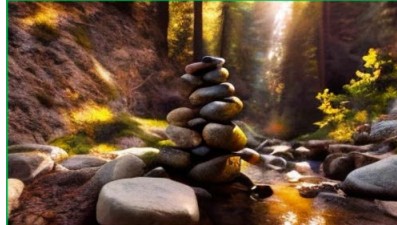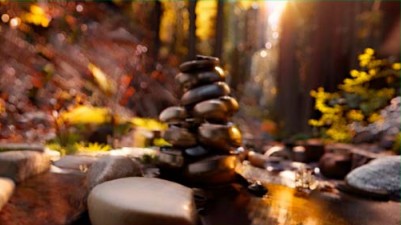

Figure 6: Post-processing sampled video clip (the left) with temporal super-resolution model (the middle) and following a spatial super-resolution model (the right). Prompts: An unstable rock cairn in the middle of a stream. (Best viewed with *Adobe Acrobat*.)

The $\mathcal{ZS}^2$ algorithm exhibits excellent adaptability across various tasks. To illustrate this, we conducted conditional generation based on ControlNet (Zhang & Agrawala, 2023), specialized generation based on DreamBooth (Ruiz et al., 2022), and implemented the Video Instruct-Pix2Pix task based on Instruct Pix2Pix (Brooks et al., 2022).

We present the corresponding results in Figure 12 and Figure 13. It is evident from these figures that our algorithm can achieve satisfactory results in a variety of task contexts.

## 7 CONCLUSION

In conclusion, this paper presents $\mathcal{ZS}^2$, a pioneering zero-shot video sampling algorithm, specifically engineered for high-quality, temporally consistent video generation. Our method, which requires no optimization or fine-tuning, can be effortlessly incorporated with a variety of image sampling techniques, thereby democratizing text-to-video generation and its associated applications. The effectiveness of our approach has been substantiated across a multitude of applications, such as conditional and specialized video generation, and instruction-guided video editing. We posit that $\mathcal{ZS}^2$ can stimulate the development of superior methods for sampling high-quality video snippets from the image diffusion model. This enhancement can be realized by merely adjusting the existing sampling algorithm, thus eliminating the necessity for any supplementary training or computational overhead.

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

## A  RELATED WORKS

**Text-to-Image Generation.**   The evolution of text-to-image synthesis began with early approaches that relied on techniques such as template-based generation (Mansimov et al., 2015) and feature matching (Reed et al., 2016), but these methods had limitations in generating realistic and diverse images. The advent of Generative Adversarial Networks (GANs) (Goodfellow et al., 2020) led to the development of deep learning-based methods for text-to-image synthesis, such as StackGAN (Zhang et al., 2017), AttnGAN (Xu et al., 2018), and MirrorGAN (Qiao et al., 2019), which enhanced image quality and diversity through innovative architectures and attention mechanisms. The advancement of transformers (Vaswani et al., 2017) further revolutionized the field, with the introduction of Dall-E (Ramesh et al., 2021), a 12-billion-parameter transformer model that introduced a two-stage training process. This was followed by the development of Parti (Yu et al., 2022), which proposed a method to generate content-rich images with multiple objects, and Make-a-Scene, which introduced a control mechanism by segmentation masks for text-to-image generation. The current state-of-the-art approaches are built upon diffusion models like GLIDE, which improved Dall-E by adding classifier-free guidance, and Dall-E 2 (Ramesh et al., 2022), which utilized the contrastive model CLIP (Radford et al., 2021) to obtain a mapping from CLIP text encodings to image encodings. Other notable models include LDM / SD (Rombach et al., 2022), which applied a diffusion model on lower-resolution encoded signals of VQ-GAN (Esser et al., 2021), and Imagen (Saharia et al., 2022), which utilized large language models for text processing. Versatile Diffusion (Xu et al., 2022) further unified text-to-image, image-to-text, and variations in a single multi-flow diffusion model. While these models have significantly improved image quality, their application in the video domain is challenging due to their probabilistic generation procedure, which makes it difficult to ensure temporal consistency.

**Text-to-Video Generation.**   Text-to-video synthesis, an emerging field of research, utilizes various methodologies for generation, often employing autoregressive transformers and diffusion processes. NUWA presents a 3D transformer encoder-decoder framework that supports both text-to-image and text-to-video generation (Wu et al., 2022a). Phenaki utilizes a bidirectional masked transformer with a causal attention mechanism, enabling the creation of arbitrarily long videos from text prompts (Villegas et al., 2022). CogVideo enhances the text-to-image model, CogView 2, by employing a multi-frame-rate hierarchical training strategy to better synchronize text and video clips  (Hong et al., 2022; Ding et al., 2022). Video Diffusion Models extend text-to-image diffusion models, training concurrently on image and video data (Ho et al., 2022b). Imagen Video establishes a cascade of video diffusion models, leveraging spatial and temporal super-resolution models to generate high-resolution, time-consistent videos (Ho et al., 2022a). Make-A-Video builds on a text-to-image synthesis model, utilizing video data in an unsupervised fashion (Singer et al., 2022). Gen-1 extends SD, proposing a structure and content-guided video editing method based on visual or textual descriptions of desired outputs (Esser et al., 2023). However, these approaches are computationally intensive and require extensive video datasets. More detrimentally, the heterogeneity of the training data between image and video datasets often leads to catastrophic forgetting of the image expert.

**Zero-shot Text-to-Video Sampling.** Recently, to mitigate the substantial computational requirements of video generation models, the concept of zero-shot text-to-video generation has been introduced, where videos are sampled directly from image generation models without any additional training. This innovative task was first introduced in the work of Tune-A-Video (Wu et al., 2022b), which proposed a one-shot video generation task by extending and tuning SD on a single reference video, albeit with training on a limited number of video sequences. Subsequent studies, such as Text2Video Zero (Khachatryan et al., 2023) and FateZero (Qi et al., 2023a), have made significant strides in this field, exploring the novel problem of zero-shot, "training-free" text-to-video synthesis. These methods build upon pre-trained text-to-image models, leveraging their superior image generation quality and extending their applicability to the video domain without additional training. However, they primarily generate brief video clips, usually consisting of a few frames, and lack effective control over content, particularly in terms of motion speed. To address these limitations, we propose $\mathcal{ZS}^2$, which uses a dependency noise sequence and temporal momentum trick to generate high-quality, more controllable long video sequences.

## B  PRELIMINARY

### B.1  DIFFUSION PROBABILISITC MODELS

Stable Diffusion (SD) is a diffusion probabilistic model that operates in the latent space of an autoencoder $\mathcal{D}(\mathcal{E}(\cdot))$, specifically VQ-GAN (Esser et al., 2021) or VQ-VAE (Van Den Oord et al., 2017), where $\mathcal{E}$ and $\mathcal{D}$ represent the encoder and decoder, respectively. More specifically, if $x_0 \in \mathbb{R}^{h \times w \times c}$ is the latent tensor of an input image $Im$ to the autoencoder, i.e., $x_0 = \mathcal{E}(Im)$, the diffusion forward process iteratively introduces Gaussian noise to the signal $x_0$:

$$q(x_t|x_{t-1}) = \mathcal{N}(x_t; \sqrt{1 - \beta_t}x_{t-1}, \beta_t I), \ t = 1, .., T \tag{3}$$

Here, $q(x_t|x_{t-1})$ is the conditional density of $x_t$ given $x_{t-1}$, and $\{\beta_t\}_{t=1}^{T}$ are hyperparameters. $T$ is selected such that the forward process completely obliterates the initial signal $x_0$, resulting in $x_T \sim \mathcal{N}(0, I)$. The objective of SD is to learn a backward process

$$p_\theta(x_{t-1}|x_t) = \mathcal{N}(x_{t-1}; \mu_\theta(x_t, t), \Sigma_\theta(x_t, t)) \tag{4}$$

for $t = T, \ldots, 1$, which enables the generation of a valid signal $x_0$ from the standard Gaussian noise $x_T$. To obtain the final image generated from $x_T$, $x_0$ is passed through the decoder of the initially chosen autoencoder: $Im = \mathcal{D}(x_0)$.

Upon mastering the aforementioned backward diffusion process as outlined in DDPM (Ho et al., 2020), a deterministic sampling process known as DDIM (Song et al., 2020a) can be utilized to establish a text-to-image synthesis framework with a textual prompt $\tau$. This can be represented by the following equation:

$$x_{t-1} = \sqrt{\alpha_{t-1}} \left( \frac{x_t - \sqrt{1 - \alpha_t}\epsilon_\theta^t(x_t, \tau)}{\sqrt{\alpha_t}} \right) + \sqrt{1 - \alpha_{t-1}}\epsilon_\theta^t(x_t, \tau), \quad t = T, \ldots, 1. \tag{5}$$

Here, $\alpha_t = \prod_{i=1}^{t}(1 - \beta_i)$ and

$$\epsilon_\theta^t(x_t, \tau) = \frac{\sqrt{1 - \alpha_t}}{\beta_t}x_t + \frac{(1 - \beta_t)(1 - \alpha_t)}{\beta_t}\mu_\theta(x_t, t, \tau). \tag{6}$$

In the context of SD, the function $\epsilon_\theta^t(x_t, \tau)$ is modeled as a neural network with a UNet-like architecture (Ronneberger et al., 2015), which is composed of convolutional and (self- and cross-) attentional blocks. The term $x_T$ is referred to as the latent code of the signal $x_0$. A method has been proposed (Dhariwal & Nichol, 2021) to apply a deterministic forward process to reconstruct the latent code $x_T$ given a signal $x_0$. For simplicity, the terms $x_t, t = 1, \ldots, T$ are also referred to as the *latent codes* of the initial signal $x_0$.

In the subsequent writing, we use superscripts in the upper right corner to denote different frames under a video sequence. For instance, $x_t^1$ represents the first frame at time $t$, while $x_T^{1:m}$ signifies the first to the $m$th frames at time $T$, and so forth.

## B.2 MUTLI-HEAD SELF ATTENTION

The attention mechanism can be characterized as a mapping function from a query and a collection of key-value pairs to an output (Vaswani et al., 2023). The input is composed of queries and keys of dimension $d_k$, and values of dimension $d_v$. The dot products of the query $Q$ with all keys $K$ are calculated, each divided by $\sqrt{d_k}$, and a softmax function is applied to derive the weights on the values $V$:

$$\text{Attention}(Q, K, V) = \text{softmax}(\frac{QK^T}{\sqrt{d_k}})V \tag{7}$$

Given that for large values of $d_k$, the dot products increase significantly in magnitude, pushing the softmax function into regions where it has extremely small gradients, it is common practice to scale the dot products by $\frac{1}{\sqrt{d_k}}$.

The multi-head attention mechanism enables the model to simultaneously attend to information from different representation subspaces at different positions. This would not be possible with a single attention head.

$$\text{MultiHead}(Q, K, V) = \text{Concat}(\text{head}_1, ..., \text{head}_\text{h})W^O$$
$$\text{where head}_\text{i} = \text{Attention}(QW_i^Q, KW_i^K, VW_i^V)$$

The projections are parameter matrices $W_i^Q \in \mathbb{R}^{d_\text{model} \times d_k}$, $W_i^K \in \mathbb{R}^{d_\text{model} \times d_k}$, $W_i^V \in \mathbb{R}^{d_\text{model} \times d_v}$ and $W^O \in \mathbb{R}^{hd_v \times d_\text{model}}$.

Self-attention is a specific instance of attention where $Q, K, V$ originate from the same input feature. Owing to its capacity to effectively extract global feature information, self-attention is extensively utilized in diffusion models.

## C PROMPTS IN TEXT TO VIDEO SYNTHESIS.

**Prompts in Figure 5** :

- An astronaut is riding a horse.
- A blue unicorn flying over a mystical land.
- Anime girl looking through a window of stars and space, sci-fi.
- Clownfish swimming through the coral reef.

**Prompts in Figure 2** :

- A man is riding a bicycle in the sunshine.
- A cat is wearing sunglasses and working as a lifeguard at a pool.
- A cute cat running in a beautiful meadow.
- A group of squirrels rowing crew.
- A beautiful girl.

**Prompts in Figure 1** :

- A bunch of colorful marbles spilling out of a red velvet bag. (Sampled from *Dreamlike Photoreal v1.0*)
- A steaming basket full of dumplings. (Sampled from *Dreamlike Photoreal v1.0*)
- Balloon full of water exploding in extreme slow motion. (Sampled from *Stable Diffusion v1.4*)
- A beautiful girl. (Sampled from *Dreamlike Photoreal v2.0*)

**Prompts in Figure 3** :

- A cat is running on the grass.
- An astronaut is skiing down the hill.
- A horse galloping on a street.

**Prompts in Figure 7** :

- A beagle in a detective's outfit.
- A bumblebee sitting on a pink flower.
- A completely destroyed car.
- A red pickup truck driving across a stream.
- A steaming hot plate piled high with spaghetti and meatballs.
- An adorable piglet in a field.
- An extravagant mansion, aerial view.
- A horse galloping on a street.
- A video showcases the beauty of nature from mountains and waterfalls to forests and oceans.
- An aerial view shows a white sandy beach on the shore of a beautiful sea.
- There is a flying through an intense battle between pirate ships in a stormy ocean.
- Balloon full of water exploding in extreme slow motion.
- An astronaut is skiing down the hill.
- An astronaut is waving his hands on the moon.

**Prompts in Figure 8** :

- A bumblebee sitting on a pink flower.
- A cat with a mullet.
- A bunny, riding a broomstick.
- A chow chow puppy.
- A goose made out of gold.
- A red convertible car with the top down.
- A tiger dressed as a doctor.
- An unstable rock cairn in the middle of a stream.
- A Ficus planted in a pot.
- A turtle is swimming in the ocean.
- A waterfall flowing through a glacier at night.
- A blue tulip.
- A chihuahua lying in a pool ring.
- A majestic sailboat.

**Prompts in Figure 9** :

- A bear dancing on Times Square.
- A beautiful girl.
- A Bigfoot is walking in a snowstorm.
- A brightly colored mushroom growing on a log.
- A bumblebee sitting on a pink flower.

- A horse is galloping through Van Gogh's "Starry Night".
- A lion reading the newspaper.
- A sheepdog running.
- A squirrel, wearing a leather jacket, riding a motorcycle, on a road made of ice.
- An unstable rock cairn in the middle of a stream.
- A teddy bear is walking down 5th Avenue, front view, beautiful sunset, close up.
- A majestic sailboat.
- There is a bird's-eye view of a highway in Los Angeles.
- There is a time-lapse of the snow land with aurora in the sky.

**Prompts in Figure 10** :

- A bumblebee sitting on a pink flower.
- A classic Packard car.
- A knight chopping wood.
- A lion reading the newspaper.
- A pirate collie dog, high resolution.
- A red convertible car with the top down.
- A red pickup truck driving across a stream.
- A robot tiger.
- A Yorkie dog eating a donut.
- An adorable piglet in a field.
- A cat riding a motorcycle.
- A koala bear is playing piano in the forest.
- A man is riding a bicycle in the sunshine.
- A toad catching a fly with its tongue.

**Prompts in Figure 11** :

- A lion reading the newspaper.
- A small cherry tomato plant in a pot with a few red tomatoes growing on it.
- A teapot shaped like an elephant head where its snout acts as the spout.
- A ficus planted in a pot.
- A monkey-rabbit hybrid.
- A shiny golden waterfall is flowing through a glacier at night.
- A teddy bear is walking down $5^{th}$ Avenue, front view, beautiful sunset, close up.
- A video showcasing the beauty of nature, from mountains and waterfalls to forests and oceans.
- A blue lobster.
- A chihuahua lying in a pool ring.
- An animated painting shows fluffy white clouds moving in the sky.
- There is a time-lapse of a fantasy landscape.
- There is a time-lapse of the snow land with aurora in the sky.
- Yellow flowers are swaying in the wind.

**Prompts in Figure 12** :

- Pose guidance: A squirrel dancing in the forests.
- Pose guidance: A bear dancing on the concrete.
- Edge guidance: A fox walking on the water surface.
- Edge guidance: A handsome man Halloween style.
- DreamBooth specialization: A beautiful girl.
- DreamBooth specialization: A handsome man.

**Prompts in Figure 13** :

- Depth guidance: Oil painting of a deer, a high-quality, detailed, and professional photo.
- Depth guidance: A dancing beautiful girl.
- Instruction Pix2Pix: Make it Expressionism style.
- Instruction Pix2Pix: Make it autumn.

**Prompts in Figure 4** :

- Pose guidance: A bear dancing on the concrete.

**Prompts in Figure 15, 14 and 16** :

- Melting ice cream dripping down the cone. (Sampled from *Stable Diffusion v1.5* with $\mu = 0.95$)

## D  IMPLEMENTATION OF DEPENDENCY NOISE MODEL AND TEMPORAL MOMENTUM ATTENTION IN PYTORCH-STYLE

```python
import torch
from typing import List

def dependency_noise_model(shape: List[int], lambda_i: float = 0.01,
    random_search_times: int = 10, linear_search_times: int = 15):
    """Dependency Noise Model for a batch.
    Args:
        shape: A list of int indicates shape as (batches, frames, ...).
        lambda_i: The hyper-parameters to control the dependency noise
    model.
            If lambda_i = 0., all frames share the same noise, and a
    still video clip will be generated.
            If lambda_i -> +inf, all frames sample independency noise, a
    random video clip will be generated.
        random_search_times: The number of times to sample in the random
    search phase.
        linear_search_times: The number of times to update in the linear
    search phase.
    Returns: A sequence of dependency noise tensors with the shape of (
    batches, frames, ...)
    """
    batches, frames, *others = shape

    def get_progressive_noise(previous_x):
        noise_x = noise = torch.randn(batches, *others)

        def compute_error(noise):
            return (torch.kl_div(noise, previous_x, log_target=True).mean
    (dim=[1, 2, 3]) - lambda_i).abs()
```

```
23
24        def compose_noise(previous, noise_x, lambda_i):
25            return torch.sqrt(lambda_i) * previous + torch.sqrt(1-
      lambda_i) * noise_x
26
27        error_bound = torch.tensor([torch.inf] * batches)
28        for _ in range(random_search_times):
29            # Coarse Random Search Phase
30            error = compute_error(noise)
31            index = error < error_bound
32            noise_x[index] = noise[index]
33            error_bound[index] = error[index]
34            noise = torch.randn(batches, *others)
35
36        alpha_i   = torch.tensor([0.1] * batches)
37        step_size = torch.tensor([0.1] * batches)
38        for _ in range(linear_search_times):
39            # Linear Search Phase
40            error = compute_error(compose_noise(previous_x, noise_x,
      alpha_i))
41            index = error <= error_bound
42            index_mask = torch.logical_not(index)
43            alpha_i[index] = alpha_i[index] + step_size[index]
44            alpha_i[index_mask] = alpha_i[index_mask] - 0.8 * step_size[
      index_mask]
45            step_size[index_mask] = step_size[index_mask] * 0.2
46            error_bound[index] = error[index]
47        return compose_noise(previous_x, noise_x, alpha_i)
48
49    noises = []
50    for index in range(frames):
51        noises.append(torch.randn(batches, *others) if index == 0 else
      get_progressive_noise(noises[-1]))
52    return torch.stack(noises, dim=1)
```

Listing 1: Implementation of dependency noise model in PyTorch-style.

```
1  import torch
2  from einops import einsum
3
4
5  def efficient_temporal_momentum_attention(x: torch.Tensor, momentum:
      float = 0.98):
6      """Efficient Temporal Momentum Attention with Matrix Operation.
7      Args:
8          x: torch.Tensor with the shape of (frames, ...).
9          momentum: The hyper-parameters to control temporal momentum
      attention.
10             If momentum = 0.0, temporal momentum attention decays to self
      -attention.
11             If momentum = 1.0, temporal momentum attention decays to
      cross-frame attention.
12      Returns: The momentum shifted tensor with the shape of (frames, ...).
13      """
14      # Build U matrix.
15      exp_mu = torch.pow(torch.tensor([momentum,] * len(x)), exponent=torch
      .arange(len(x)))
16      exp_mu_matrix = torch.stack([torch.roll(exp_mu, i) for i in range(len
      (exp_mu))]).T
17      U = torch.tril(exp_mu_matrix)
18
19      # Matrix multiply
20      x[1:] = x[1:] * (1 - momentum)
21
22      return einsum("ff, fbnc -> fbnc", U, x)
```

```python
23
24 def temporal_momentum_attention(x: torch.Tensor, momentum: float = 0.98):
25     """Momentum Attention with `for` loop.
26     Args:
27         x: torch.Tensor with the shape of (frames, ...).
28         momentum: The hyper-parameters to control temporal momentum
     attention.
29             If momentum = 0.0, temporal momentum attention decays to self
     -attention.
30             If momentum = 1.0, temporal momentum attention decays to
     cross-frame attention.
31     Returns: The momentum shifted tensor with the shape of (frames, ...).
32     """
33     return torch.stack([x[index] if index == 0 else x[index-1] * momentum
      + x[index] * (1-momentum) for index in range(len(x))])
```

Listing 2: Implementation of temporal momentum attention in PyTorch-style.

## E    MORE TEXT TO VIDEO SAMPLING RESULTS.

Figure 7, 8, 9, 10 and 11 demonstrate more video clips sampled from Stable Diffusion v1.4 and v1.5, Dreamlike Photoreal v1.0 and v2.0, and Openjourney with $\mathcal{ZS}^2$ respectively.

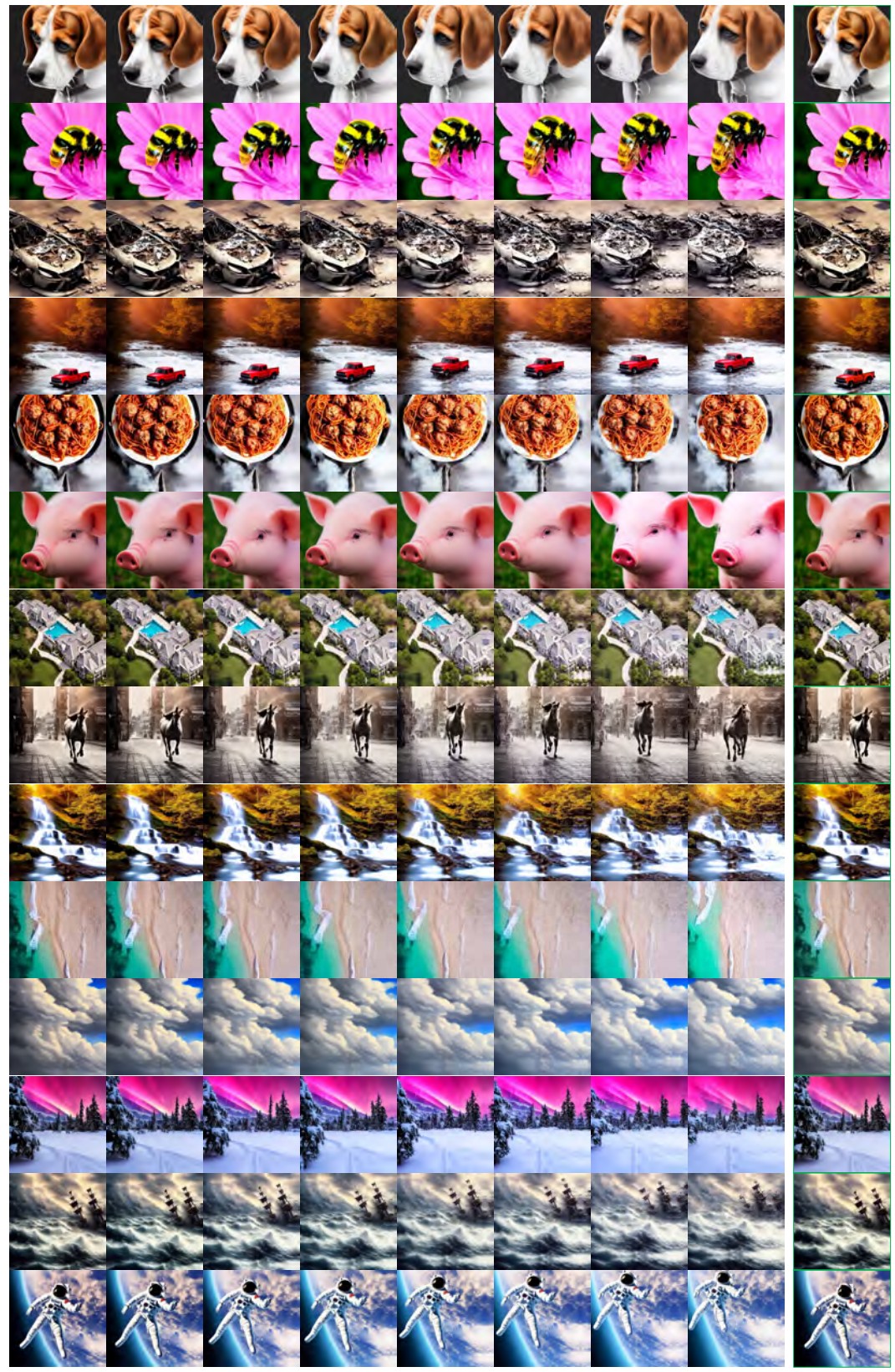

Figure 7: More results sampled from Stable-Diffusion v1.4.

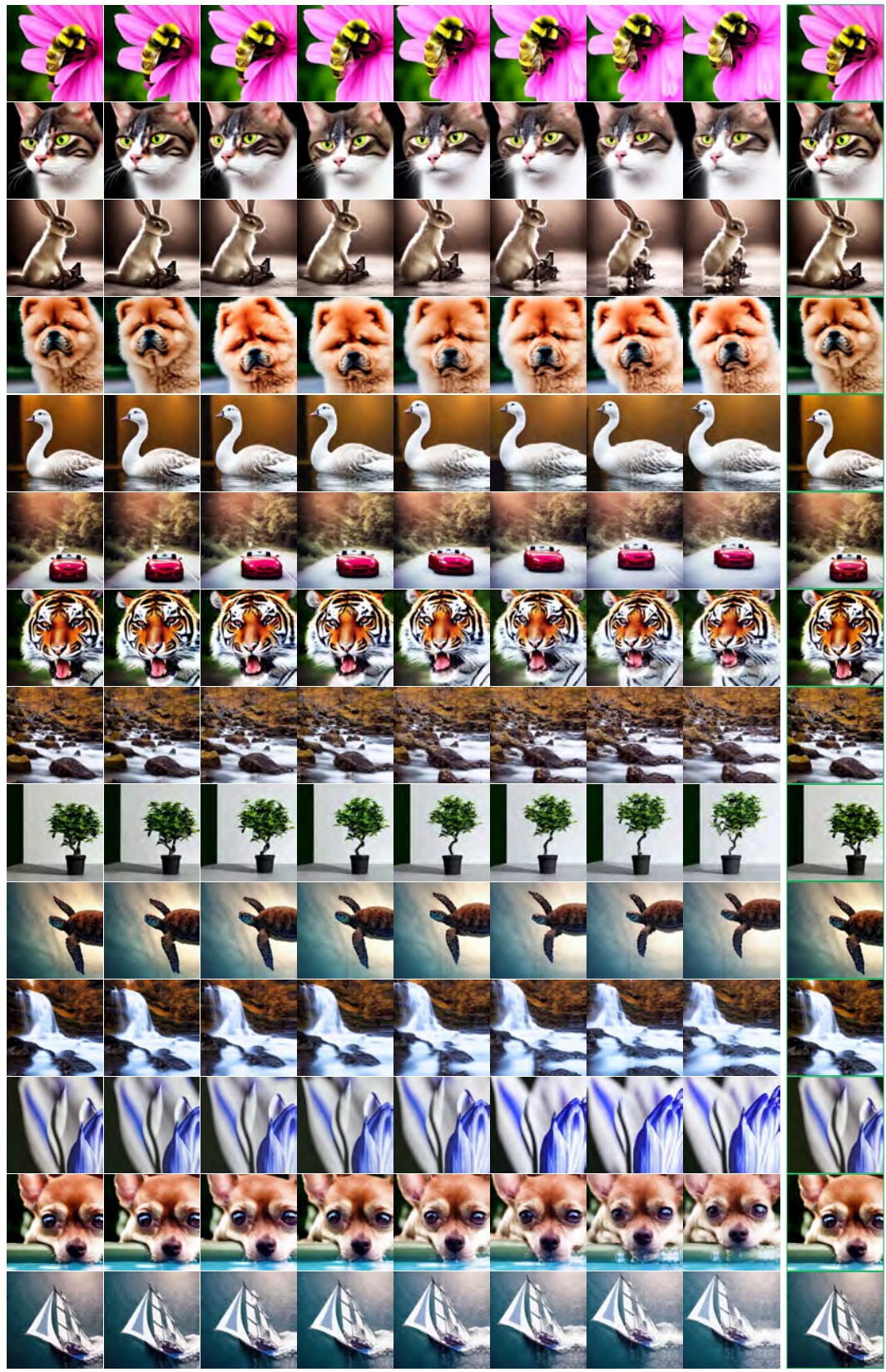

Figure 8: More results sampled from Stable-Diffusion v1.5.

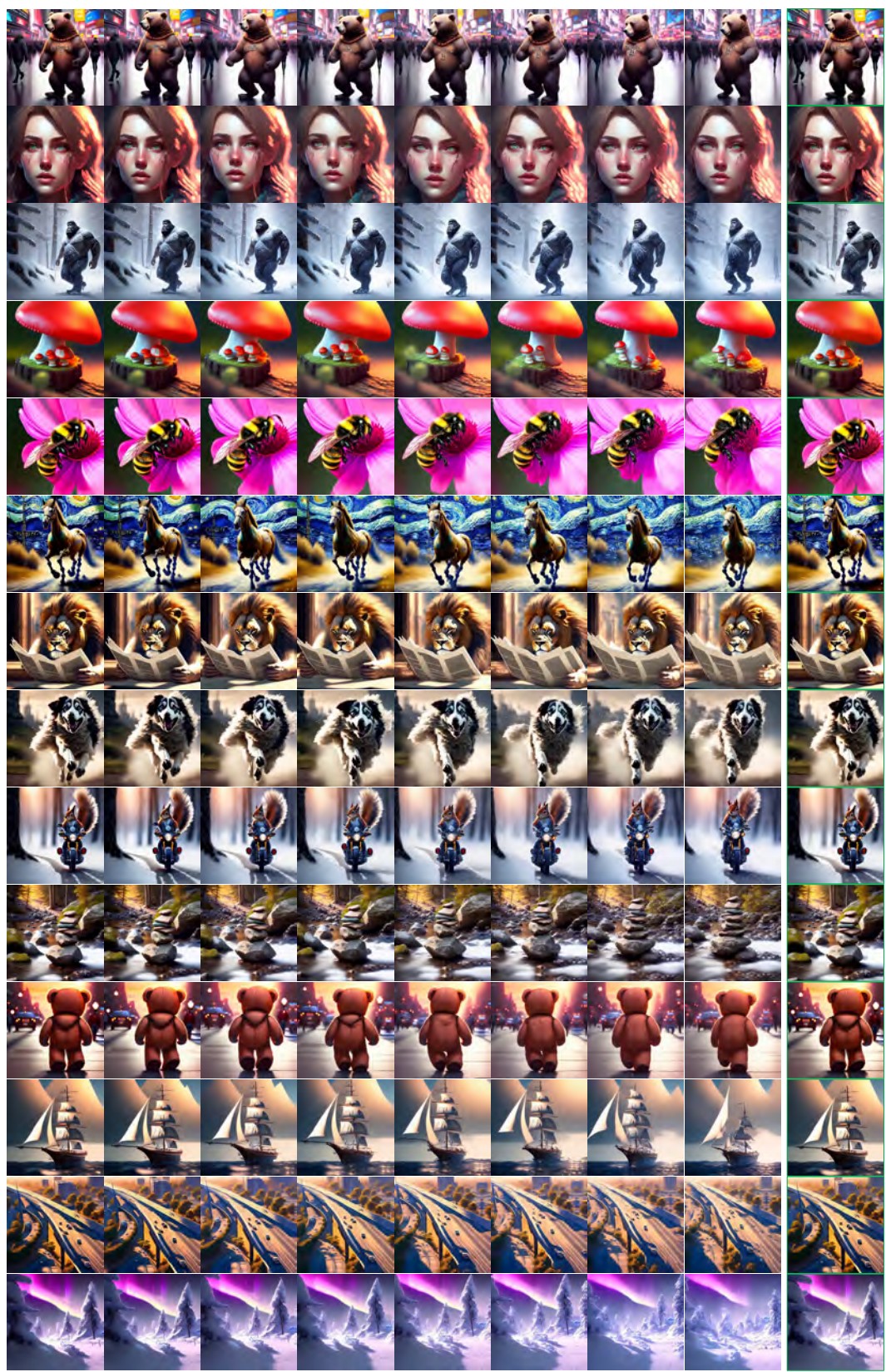

Figure 9: More results sampled from Dreamlike Photoreal v1.0.

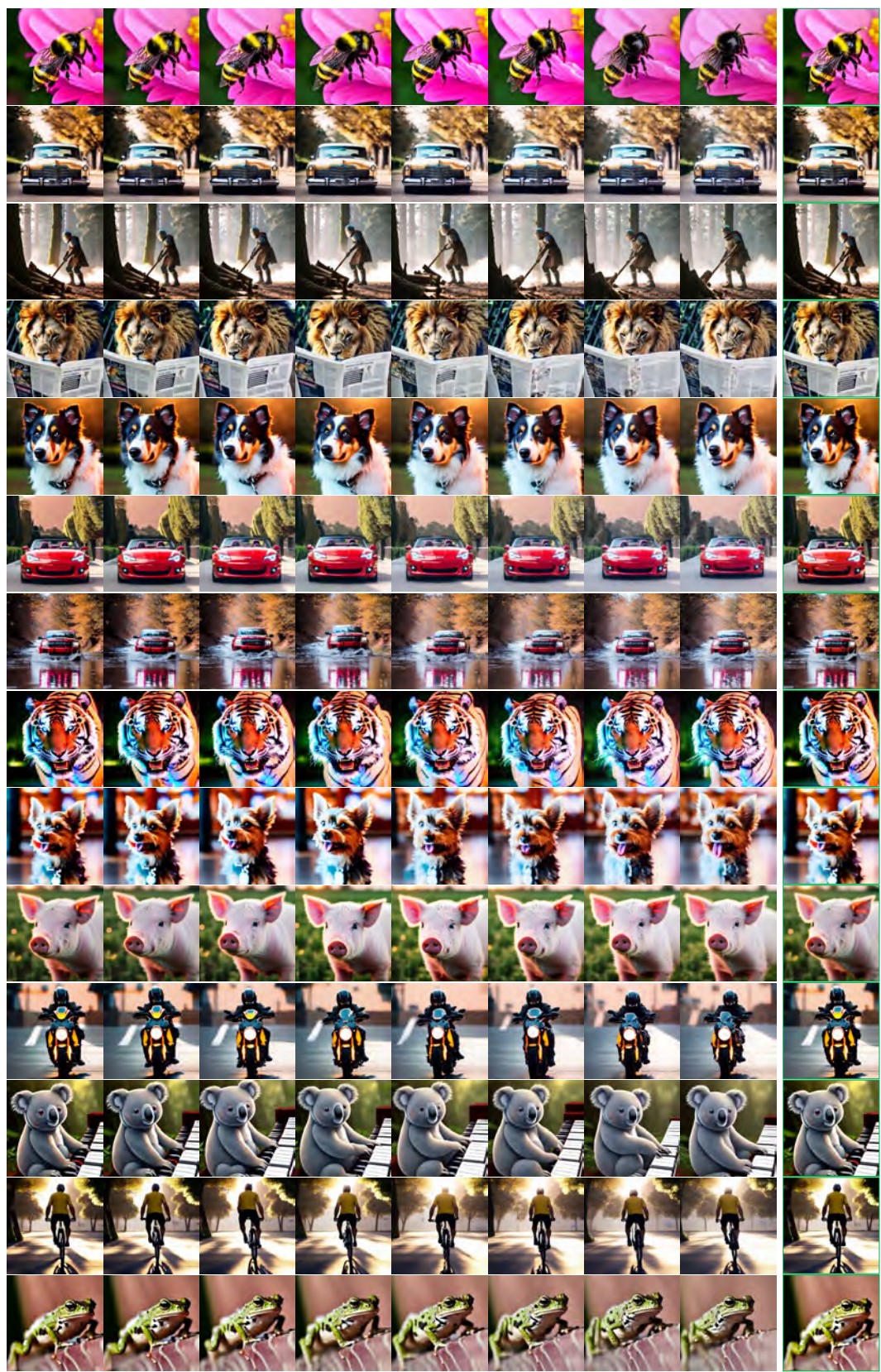

Figure 10: More results sampled from Dreamlike Photoreal v2.0.

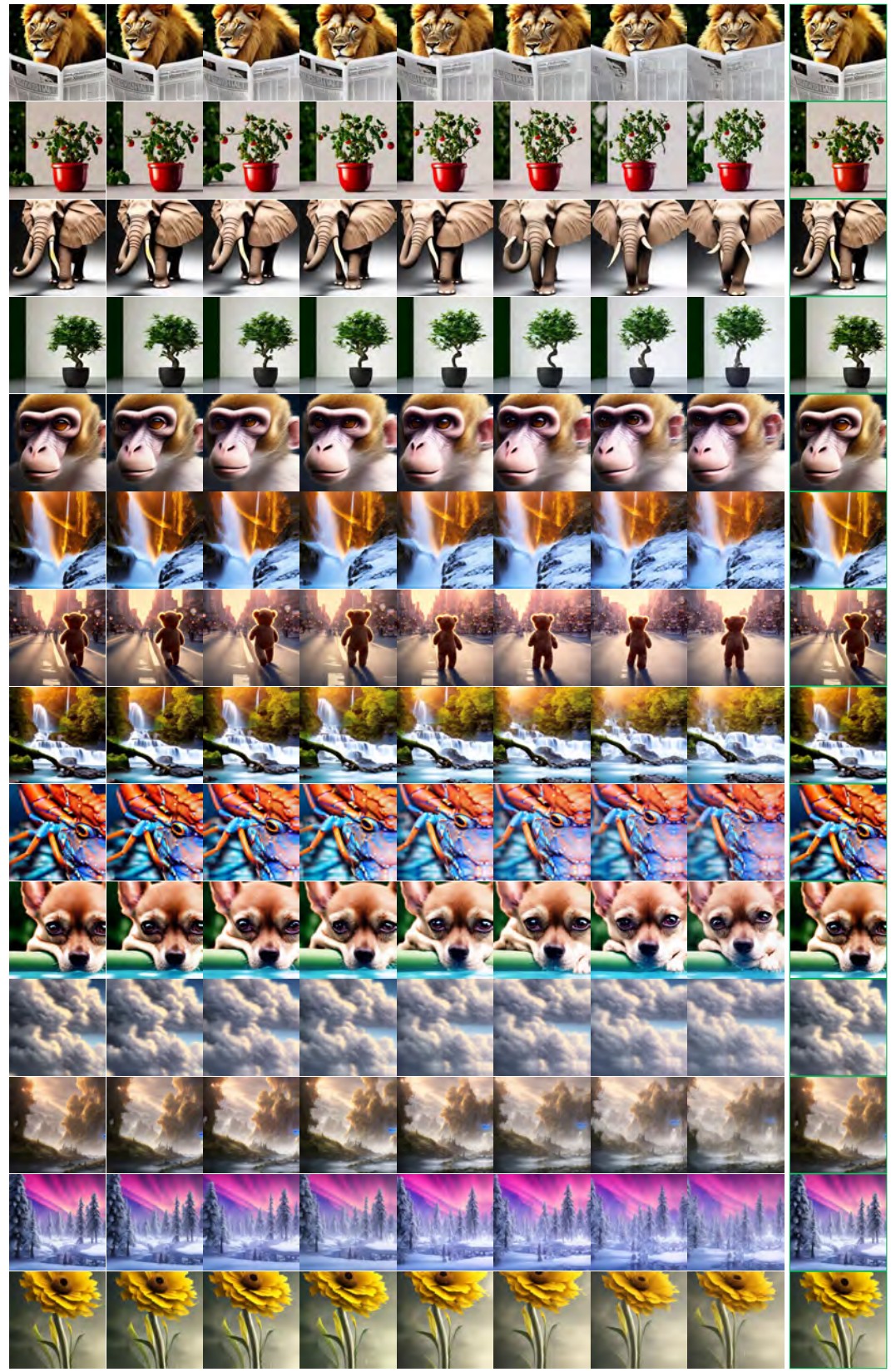

Figure 11: More results sampled from Open Journey.

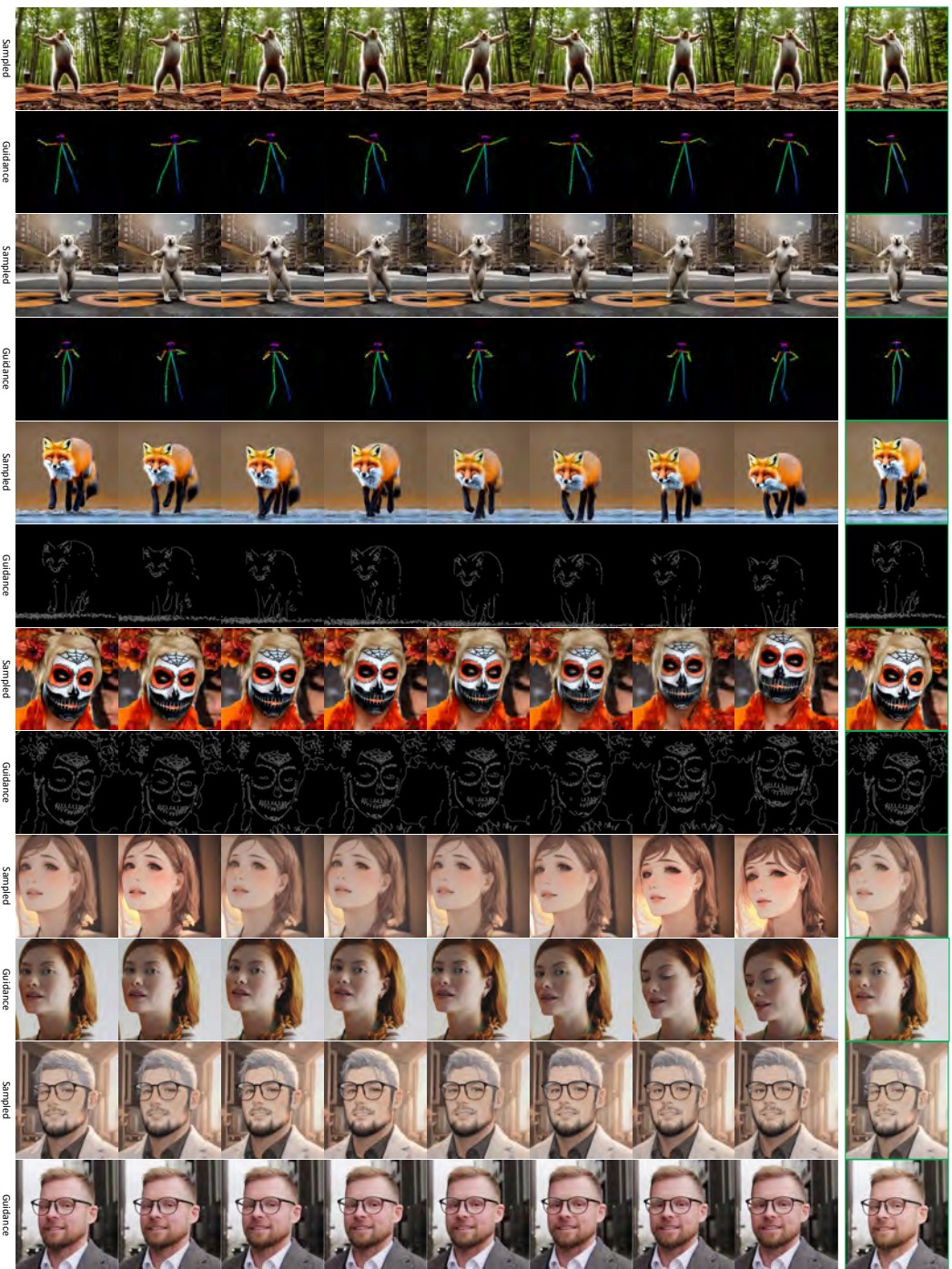

Figure 12: Performace on extensions tasks.

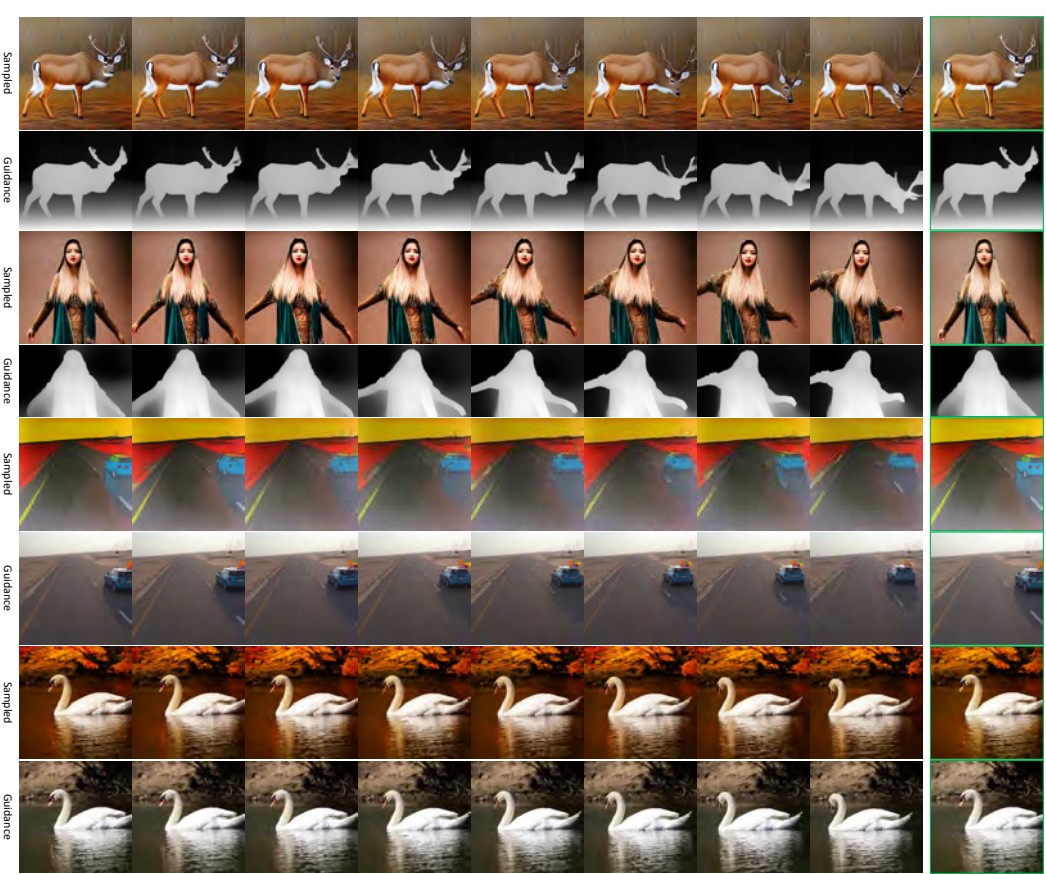

Figure 13: Performance on extensions tasks.

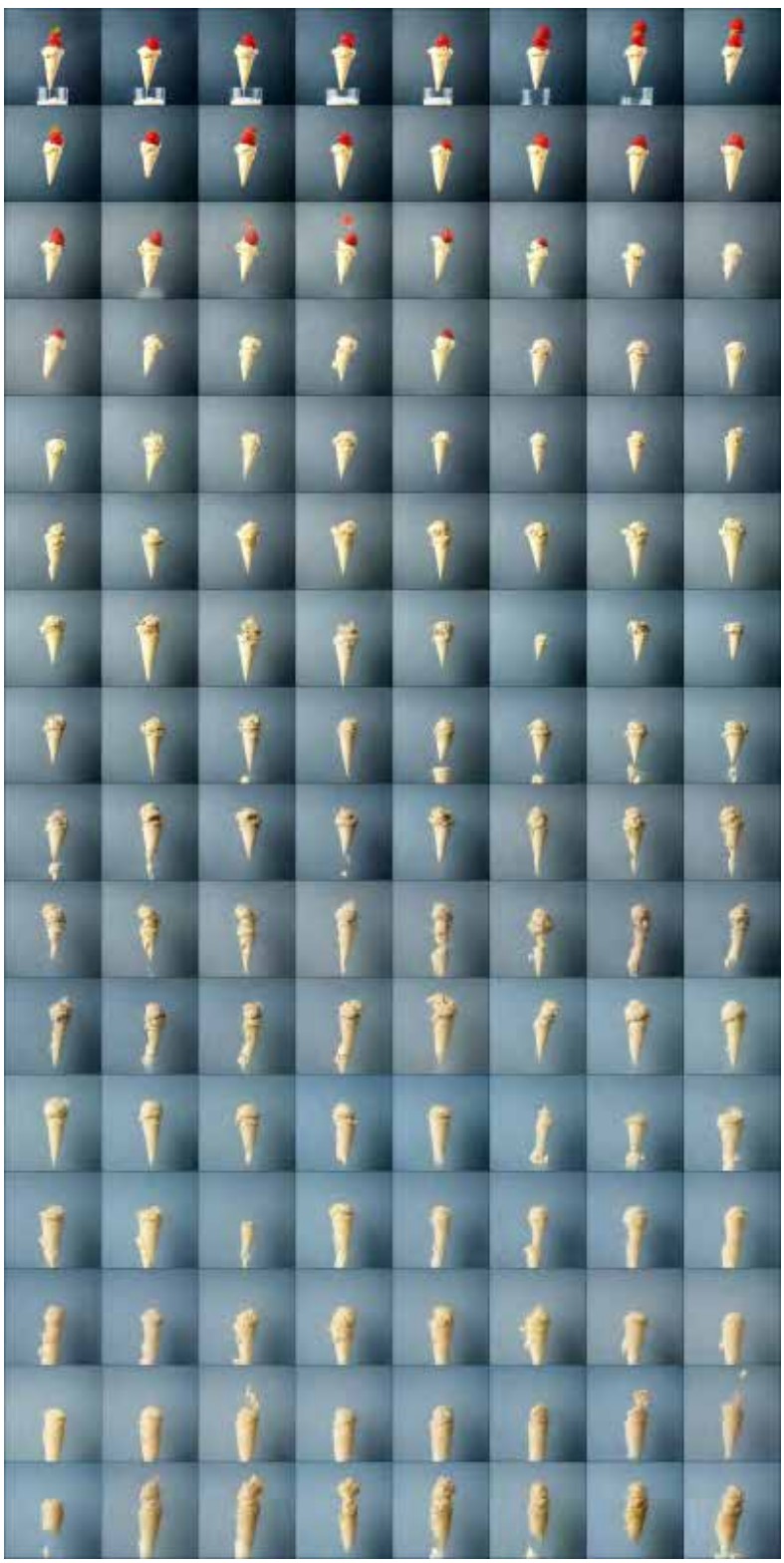

Figure 14: The progressive noise model is applied to sample a sequence of frames, proceeding from left to right and top to bottom. Although high-quality images can be initially sampled, the semantic information tends to rapidly decay due to the accumulation of noise across frames. This leads to an inability to sample high-quality video sequences.

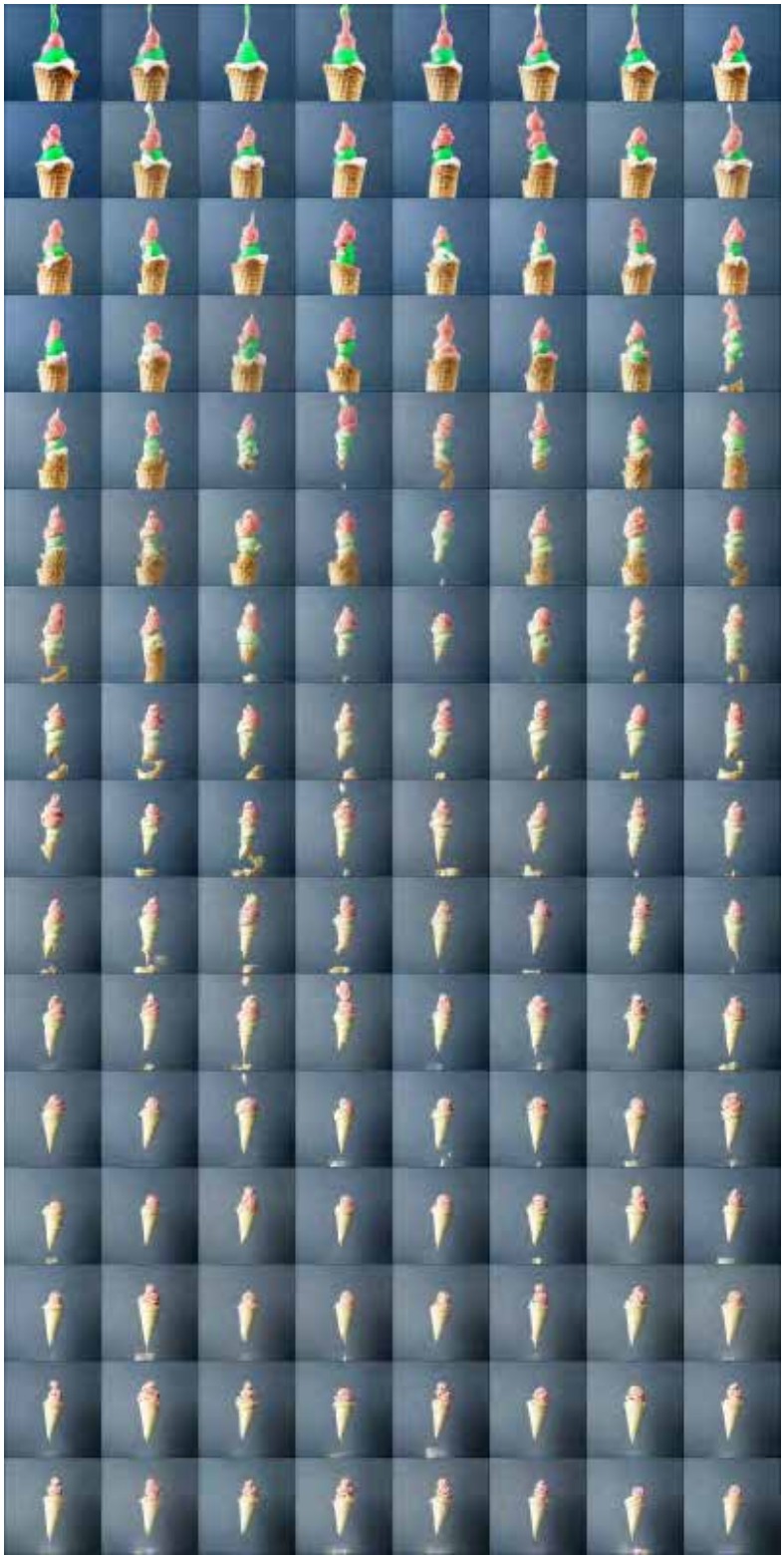

Figure 15: The mixed noise model is applied to sample a sequence of frames, proceeding from left to right and top to bottom. Although high-quality images can be initially sampled, the semantic information tends to rapidly decay due to the accumulation of noise across frames. This leads to an inability to sample high-quality video sequences.

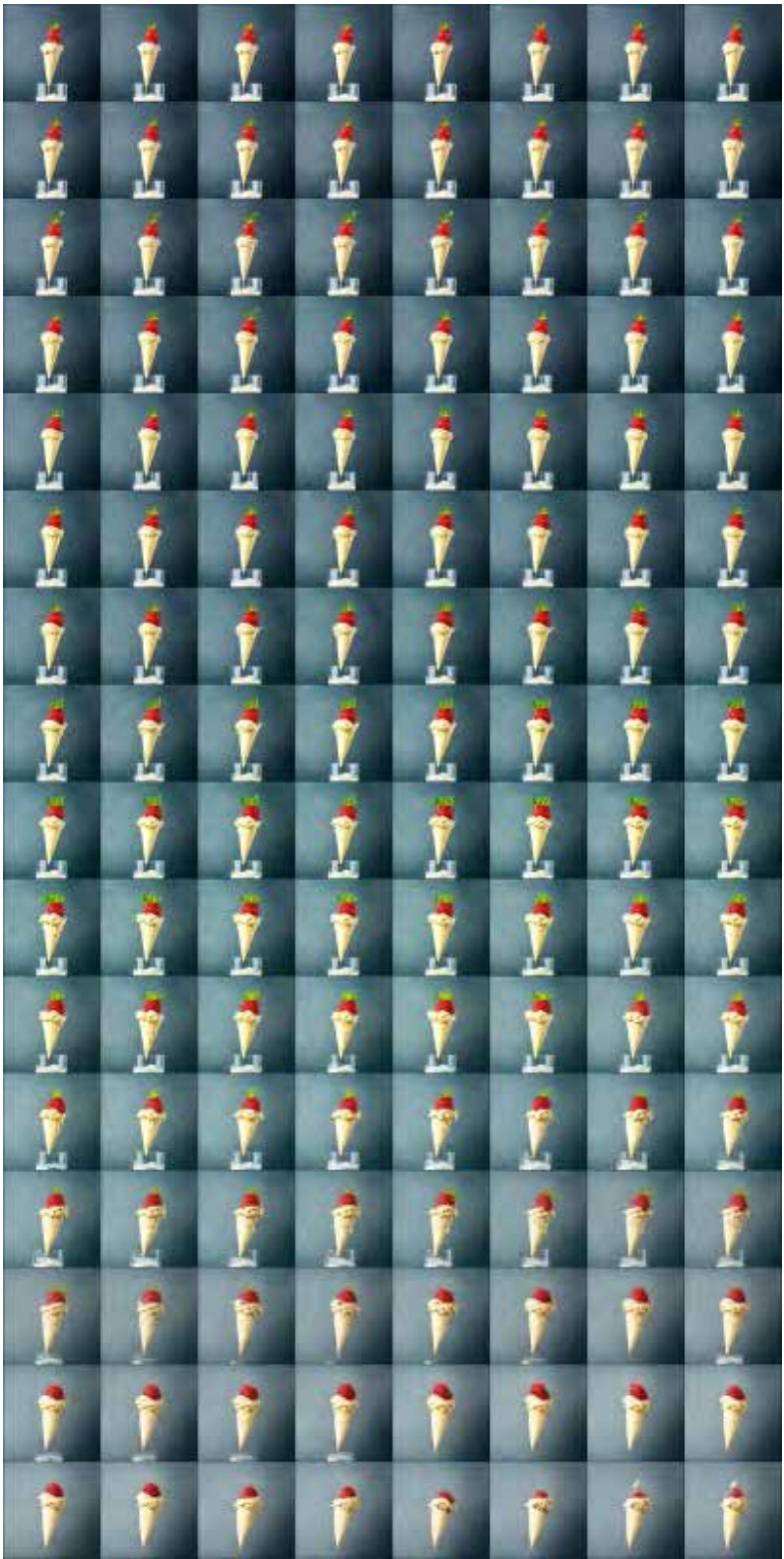

Figure 16: The dependency noise model is employed to sample a sequence of frames, proceeding from left to right and top to bottom. Unlike existing noise models, i.e., the progressive and mixed noise models, the proposed noise model is capable of preserving semantic information more effectively over long sequences, making it more suitable for zero-shot sampling.

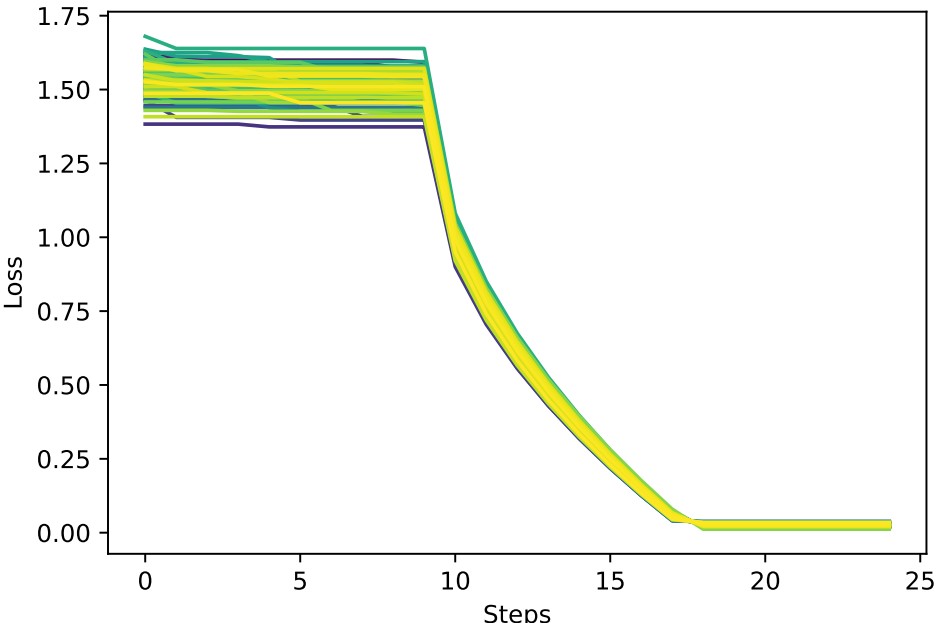

Figure 17: The convergence speed of our proposed two-stage algorithm is noteworthy. In the first stage (initial 10 steps), we primarily sample random noises and select the most suitable one as the initial noise for the second stage. In the second stage (subsequent 15 steps), we are able to converge rapidly to a minimal error. This is mainly attributed to the linear additivity of Gaussian noise, which allows us to quickly search for $\alpha$ between $0 \rightarrow 1$. In our experiments, we found that although the random sampling in the first stage does not yield satisfactory noise, it significantly enriches the diversity of the final generated video clips. Without the random sampling in the first stage, the content of the generated videos tends to be more monotonous.

