# OpenReview forum: "Zero-Shot Video Sampling from Image"
_ICLR.cc/2024/Conference — ICLR 2024 Conference Withdrawn Submission_

### Official Review · Reviewer_b1a9 · 2023-10-22

**Soundness:** 2 fair
**Presentation:** 2 fair
**Contribution:** 2 fair
**Rating:** 5
**Confidence:** 3

**Summary:**

This work proposes a Zero-Shot video Sampling algorithm that can directly sampling high-quality video clips from existing image synthesis methods without further training or optimization. Concretely, the authors propose to utilize the dependency noise model and temporal momentum attention to ensure content consistency and animation coherence. Empirical validations demonstrate the effectiveness of the proposed method.

**Strengths:**

1. Zero-shot generalization ability. The proposed method utilized the pre-trained image synthesis methods and can be generalized to video domain tasks without any training operation. This is very helpful for the application on edge devices and resource-limited users.
2. The writing is clear. Overall, the authors present their method clearly and the whole paper is easy to understand.
3. The visualization looks promising compared to baseline methods.

**Weaknesses:**

1. Missing comparisons. The authors mentioned in the introduction that ' Text2Video Zero (Khachatryan et al., 2023) and FateZero (Qi et al., 2023a) have made progress in exploring the novel problem of zero-shot, ”training-free” video synthesis.', but the proposed method only compare with Text2Video Zero and the comparisons with FateZero is missing. Further, in the Text2Video Zero paper, they compare with CogVideo, which is also ignored in this work.
2. Missing ablation. The authors didn't provide any ablation study in the paper which is very important to demonstrate the effectiveness of each design.
3. Organization needs to be improved. Some important figures should be put in the main text, such as Figure 16/17. Moreover, most of the figures should be put near the text where they are mentioned. Also, the caption in front of the figures are too small.

**Questions:**

Besides the questions mentioned above, the reviewer has one more question: how do the authors define the content consistency or animation coherence? In the given 8-frame video examples, content consistency can also be interpreted as a lack of diversity. Previous methods exhibit more diversity between video frames, but is this still a problem when applying their method to generate longer video clips? Or does the proposed method still perform well on longer video generation?

---

### Official Review · Reviewer_m7w3 · 2023-10-26

**Soundness:** 3 good
**Presentation:** 3 good
**Contribution:** 3 good
**Rating:** 6
**Confidence:** 4

**Summary:**

The paper presents a novel technique for utilizing the existing SOTA text-image baselines and performing the task of text-video generation. The author introduces an optimizing constraint over the noise vector to ensure temporal consistency across the generated frames. Authors also introduced temporal attention momentum module which ensures that the current frame has context of neighbouring frames. Video results have been provided by the authors to evaluate the approach.

**Strengths:**

* Video results are temporarily more coherent than their counterparts.
* The method is empirically more effective in utilizing the foundational models.

**Weaknesses:**

* The resolution of the output video sequence seems to be 64x64. This seems low resolution. Is the limitation coming from the foundational model, or is there some other issue ?
* The number of frames shown per video seems pretty low. My question is, when is the fidelity of the output wane.
* What are the limitations of the method, as there is no section in the paper summarizing them?
* The results mainly consist of camera motions.

**Questions:**

* How was the direction of KL divergence determined in Eqn 1?

---

### Official Review · Reviewer_5J99 · 2023-10-31

**Soundness:** 2 fair
**Presentation:** 1 poor
**Contribution:** 2 fair
**Rating:** 3
**Confidence:** 4

**Summary:**

- This paper proposed a training-free zero-shot video diffusion method. It can be applied to any image diffusion model. The core of its method is a dependency noise model, which tries to minimize the KL divergence between adjacent frames. To restrict the initial noise not to deviate significantly from the normal distribution, the author only optimizes the KL divergence with line search. The second component of the method is momentum attention, which replaces the Key and Value in 2D attention with the temporal momentum of the keys and values.
- The author showed most of the video generation results only with a few frames(8 frames), and a few with 128 frames. All the results are shown with low resolutions. For experimental comparison, the author compared with Text2Video-Zero and direct random noise sampling only using CLIP-scores (which does not make sense for video generation), and showed two sets of visual comparisons with other state-of-the-art video generations.

**Strengths:**

- Training-free, zero-shot video generation is a big strength of the proposed method. Also, the proposed noise sampling is efficient.
- The visual quality of the generated video seems fair, but higher-resolution results and more quantitative comparisons are needed to justify this further.

**Weaknesses:**

- The biggest weakness is in the experiment evaluation part. I cannot get a clear sense of the quality of the generated videos.
    - No significant quantitative comparison is performed. Only one set of clip scores is reported, and clip score is not a good metric for video generation!(It does not consider temporal consistency) I recommend the author add more metrics like Frechet Video Distance (FVD) (Unterthiner et al., 2018), and Kernel Video Distance (KVD) (Unterthiner et al., 2019).
    - The proposed algorithm is training-free, so it can easily generate high-resolution videos using high-resolution text-to-image models like SDXL. It would be great for the author to add more high-resolution results.
    - For visual comparison, adding more videos, especially videos with higher frame rates in supplementary materials or external websites, is very helpful. Also user study is welcomed for more solid evaluations.
- The presentation has some weaknesses.
    - The first paragraph of section 2: "Present text-to-video synthesis techniques either require costly training on large-scale text-video paired data ...., or necessitate fine-tuning on a reference video...". This sentence is logically wrong since the author cited some training-free video generation methods, like the text2video-zero.
    - In figures 2,3,4,5... The font in the figure is too small. It would be better to improve the font size.
- Ablations of the two proposed components is missing. It would be interesting to see what the videos will be like without the momentum attention, or without the dependency noise sampling.

**Questions:**

I list most of my suggestions for the experment section in the above Weaknesses part.

---

### Official Review · Reviewer_oRp9 · 2023-11-01

**Soundness:** 1 poor
**Presentation:** 2 fair
**Contribution:** 2 fair
**Rating:** 3
**Confidence:** 5

**Summary:**

This paper proposes an approach for generating smooth videos from trained text-to-image model without finetuning. The approach contains two parts, one is about constructing noise across frames that stays relatively consistent; the other is about changing the self-attention layers of 2D U-Net to cross-frame attention accumulated over time. Results show that the method can generate reasonable moving videos.

**Strengths:**

The idea of zero-shot video generation with pretrained image generation is worthwhile to explore, which can maximally maintain and leverage the generation ability of the 2D model.

**Weaknesses:**

The paper contains a lot of mathematical flaws and does not show convincing empirical results. Specifically,

- Sec 3.1 is not technically sound. The KL divergence between two even independent standard Gaussian distributions would be zero, not like the paper mentioned that $\lambda_i$ will converge to infinity. It doesn't make sense to control the similarity between $\epsilon^i$ and $\epsilon^{i-1}$ by computing the KL divergence between the two distributions.

- The end of page 4: "revisiting Eq. 1,..., from the definition of KL divergence" is wrong. One can not derive one variable from the other by knowing the KL divergence between them.

- For all qualitative results, there are a separate column in the right end, which seemingly tries to show image generation results? I don't know what that column stands for.

- It doesn't indicate the video quality by computing the CLIP score and comparing with image models. Better metric would be e.g., FVD.

- It is difficult to compare with other baseline approaches by only checking Figure 5, which generates videos with completely different text prompts, random seeds and resolutions.

**Questions:**

See comments above